# Non-Alcoholic Fatty Liver Disease and Liver Fibrosis in Persons with Type 2 Diabetes Mellitus in Ghana: A Study of Prevalence, Severity, and Contributing Factors Using Transient Elastography

**DOI:** 10.3390/jcm12113741

**Published:** 2023-05-29

**Authors:** Yaw Amo Wiafe, Mary Yeboah Afihene, Enoch Odame Anto, Richmond Ashitey Nmai, Lois Amoah-Kumi, Joseph Frimpong, Francis D. Dickson, Samuel O. Antwi, Lewis R. Roberts

**Affiliations:** 1Department of Medical Diagnostics, College of Health Sciences, Kwame Nkrumah University of Science and Technology, Kumasi, Ghana; eoanto@knust.edu.gh (E.O.A.); dostydadson@gmail.com (R.A.N.); lamoahkumi@gmail.com (L.A.-K.); jfrimps122@gmail.com (J.F.); 2Department of Medicine, College of Health Sciences, Kwame Nkrumah University of Science and Technology, Kumasi, Ghana; maryafihene@gmail.com; 3Nova Surgery Center, Accra, Ghana; francisddickson@gmail.com; 4Division of Epidemiology, Department of Quantitative Health Sciences, Mayo Clinic College of Medicine and Science, Jacksonville, FL 32224, USA; antwi.samuel@mayo.edu; 5Division of Gastroenterology and Hepatology, Mayo Clinic College of Medicine and Science, Rochester, MN 55905, USA; roberts.lewis@mayo.edu

**Keywords:** non-alcoholic fatty liver disease, type 2 diabetes mellitus, prevalence, weight management, liver fibrosis, controlled attenuation parameter score

## Abstract

Type 2 diabetes mellitus (T2DM) is a metabolic disorder characterized by hyperglycemia, insulin resistance, and pancreatic islet cell dysfunction. T2DM is associated with non-alcoholic fatty liver disease (NAFLD) because of impaired glucose metabolism in both conditions. However, it is widely assumed that people with T2DM in sub-Saharan Africa (SSA) have a lower prevalence of NAFLD than in other parts of the world. With our recent access to transient elastography, we aimed to investigate the prevalence of, severity of, and contributing factors to NAFLD in persons with T2DM in Ghana. We performed a cross-sectional study recruiting 218 individuals with T2DM at the Kwadaso Seventh-Day Adventist and Mount Sinai Hospitals in the Ashanti region of Ghana using a simple randomized sampling technique. A structured questionnaire was used to obtain socio-demographic information, clinical history, exercise and other lifestyle factors, and anthropometric measurements. Transient elastography was performed using a FibroScan^®^ machine to obtain the Controlled Attenuation Parameter (CAP) score and liver fibrosis score. The prevalence of NAFLD among Ghanaian T2DM participants was 51.4% (112/218), of whom 11.6% had significant liver fibrosis. An evaluation of the NAFLD group (*n* = 112) versus the non-NAFLD group (*n* = 106) revealed a higher BMI (28.7 vs. 25.2 kg/m^2^, *p* = 0.001), waist circumference (106.0 vs. 98.0 cm, *p* = 0.001), hip circumference (107.0 vs. 100.5 cm, *p* = 0.003), and waist-to-height ratio (0.66 vs. 0.62, *p* = 0.001) in T2DM patients with NAFLD compared to those without NAFLD. Being obese was an independent predictor of NAFLD in persons with T2DM than known history of hypertension and dyslipidaemia.

## 1. Introduction

Non-alcoholic fatty liver disease (NAFLD) is the leading cause of chronic liver disease in the western world, and is being increasingly recognized as the major cause globally [1]. People with type 2 diabetes mellitus (T2DM) have a higher prevalence of NAFLD than the general population, because of the association between insulin resistance and excess fat accumulation [2]. Globally, NAFLD is reported to exist in 25% of the general population [3,4,5] and over 70% of people with T2DM [6,7].

Existing data suggest that western countries are the most affected by NAFLD, followed by Middle Eastern countries, with African countries being the least affected [7]. However, it is also believed that the NAFLD prevalence in Africa has been underestimated [3,8]. The need for more population-based and clinical studies from Africa has therefore been recommended [8].

While NAFLD does not progress to cirrhosis in most people, those with T2DM have a higher risk of progression to steatohepatitis and cirrhosis [9,10]. Consequently, recent guidelines recommend that people with T2DM should be tested regularly for NAFLD and advanced fibrosis [10,11]. The severity of hepatic fibrosis is linked to long-term effects on the liver [12]. In primary care clinics, especially for people who have additional metabolic conditions such as hypertension and hyperlipidemia, the risk of advanced fibrosis should be assessed. [13]. Transient elastography is one of the suggested methods for assessing advanced fibrosis [13]. However, in Africa, not only are the data on NAFLD scarce, but there is also a lack of available data about the levels of liver fibrosis in persons with co-existing T2DM and NAFLD.

This study was therefore conducted in two primary care hospitals in Ghana to determine the prevalence of, severity of, and factors contributing to the co-existence of T2DM and NAFLD. According to the International Diabetes Federation (IDF), the prevalence of T2DM in Africa is <5%–≥7% [14]. A recent systematic review and meta-analysis estimated the prevalence of T2DM in Ghana as 6.46% [15].

## 2. Materials and Methods

This prospective cross-sectional study was carried out at the Kwadaso Seventh-Day Adventist and Mount Sinai Hospitals in Kumasi, Ghana. The two primary care hospitals are 15 km apart on the Kumasi–Sunyani highway. The Kwadaso Seventh-Day Adventist hospital is mainly used by residents of the Kwadaso-Asuoyeboa community as their primary healthcare center, whilst residents of Abuakwa-Koforidua primarily use the Mount Sinai Hospital. Diabetes clinics are available at both hospitals for residents of the two communities.

All outpatients who visited the diabetes clinic in September 2022 were invited to participate in the study. We excluded patients with type 1 diabetes, other hyperglycemic conditions that could not be confirmed as type 2 diabetes, and those who had chronic hepatitis B or C. A total of 218 T2DM patients comprising 114 individuals from Kwadaso Seventh-Day Adventist hospital and 104 individuals from Mount Sinai Hospital were recruited using a simple randomized sampling technique.

The study was conducted in accordance with the Declaration of Helsinki, and approved by the Committee on Human Research Publication and Ethics (CHRPE) of the Kwame Nkrumah University of Science and Technology, Kumasi, Ghana. Written informed consent was obtained from all participants.

A structured questionnaire was used to obtain socio-demographic information, disease history, exercise, and lifestyle history. Physical measurements were obtained from each participant, including systolic blood pressure (SBP) and diastolic blood pressure (DBP) measured using a standard sphygmomanometer (Omron HEM-711DLX, Kyoto, Japan). The average value for the two measurements (with a 5-min interval between measurements) was recorded to the nearest 2.0 mmHg.

Weight and height were measured using a weighing scale and stadiometer and body mass index (BMI) was calculated as BMI = weight (kg)/height (m)^2^. Waist circumference (to the nearest 0.1 cm) was measured with a Gulick II spring-loaded measuring tape (Gay Mills, WI) halfway between the inferior angles of the ribs and the suprailiac crests. The hip circumference was measured at the widest diameter around the gluteal protuberance to the nearest 0.1 cm. Waist-to-hip ratio (WHR) was calculated as WHR = waist (cm)/hip (cm) whereas waist-to-height ratio was calculated as WHtR = waist (cm)/height (cm).

Transient elastography was performed using a FibroScan^®^ machine (Echosens^®^, Paris, France). The machine was equipped with the M and XL probes. Participants in the study were assessed in either the supine or lateral decubitus position on an examination bed. For all participants, the probe was positioned at one of the intercostal spaces of the right hypochondriac region. By pressing a button on the probe, a sound wave was transmitted into the body of the patient which was felt by the patient as a mild vibration. An indicator on the FibroScan^®^ machine’s monitor gave feedback as to whether the sound transmission was successful. After several cycles of acoustic transmission, the Controlled Attenuation Parameter (CAP) score of the patient was calculated by the machine in decibels per meter (dB/m). The machine also estimated the liver fibrosis score, which was presented as median in kilopascals (kPa).

Classification of the degree of steatosis was carried out in accordance with a previous publication [16]. A CAP score of <248 dB/m referred to **S0** (i.e., no steatosis); 248–267 dB/m referred to **S1** (i.e., mild steatosis); >268–279 dB/m referred to **S2** (i.e., moderate steatosis); and >280 dB/m referred to **S3** (i.e., severe steatosis) [15]. The classifications of levels of fibrosis used <7.0 kPa as the cut-off for normal (i.e., F0–F1) as published [17]. A stiffness of >7.0 kPa refers to increasing fibrosis, with >7.9 kPa to 8.7 kPa consistent with histologic F2, 8.8 to <11.7 kPa consistent with F3 fibrosis, and ≥11.7 kPa consistent with advanced F4 fibrosis [17].

Data were entered into Microsoft Excel 2016 and analyses performed using Jamovi version 2.3.18 (jamovi.org, Sydney, Australia) and GraphPad Prism 8.0.1 (GraphPad LLC, San Diego, CA, USA). Categorical data were presented as frequency (proportion). The normality of continuous variables was tested using a Kolmogorov–Smirnov test. Non-parametric data were presented as median (interquartile range). Differences between groups were tested for significance using a Mann–Whitney U-test for non-parametric data (for continuous variables) or a chi-squared or Fisher exact test (for categorical variables). The association of anthropometric and metabolic variables with non-alcoholic fatty liver disease was analyzed by multinomial logistic regression and linear logistic regression. Statistical significance was set at *p* < 0.05.

## 3. Results

### 3.1. Prevalence of NAFLD in T2DM

The overall prevalence of NAFLD among the subjects with T2DM was 51.4% (112/218). As shown in Figure 1, the frequency of NAFLD in persons with T2DM was highest among subjects with a BMI ≥ 30 kg/m^2^ (obese group; 70.6%), followed by those who were overweight (BMI between 25 and 29.9 kg/m^2^; 53.4%) and then subjects with normal weight (BMI < 25 kg/m^2^; 33.8%) (Figure 1). The prevalence of NAFLD was remarkably higher in patients in their 50s and 60s (Table 1).

### 3.2. Demographic, Anthropometric, Lifestyle and Clinical Characteristics of T2DM with and without NAFLD

Table 1 summarizes the key demographic, anthropometric, lifestyle and clinical characteristics of the T2DM patients with and without NAFLD. Significantly higher BMI (28.74 vs. 25.20 kg/m^2^, *p* = 0.001), waist circumference (106.00 vs. 98.00 cm, *p* = 0.001), hip circumference (107.00 vs. 100.50 cm, *p* = 0.003), waist-to-height ratio (0.66 vs. 0.62, *p* = 0.001), and CAP score (278.00 vs. 218.00 dB/m, *p* < 0.0001) were observed in T2DM with NAFLD compared to those without NAFLD. However, waist-to-hip ratio (*p* = 0.340), systolic blood pressure (*p* = 0.857), and diastolic pressure (*p* = 0.996) did not vary significantly between the groups. There were no significant differences in age (*p* = 0.671) or duration of diabetes (*p* = 0.344) between the NAFLD and non-NAFLD groups. The difference in gender between participants with NAFLD and those without NAFLD approached significance (54.7% of females vs. 39.6% of males with NAFLD, *p* = 0.064). Hypertension (*p* = 0.542), dyslipidaemia (*p* = 0.130), current smoking (*p* = 0.608), and daily physical activity (*p* = 0.756) were not significantly different between participants with NAFLD and those without NAFLD. Smoking rates were very low, at <2% in both groups.

### 3.3. Predictive Risk Factors of NAFLD

Table 2 shows a multinomial logistic regression analysis of the factors predictive of NAFLD in patients with T2DM. Body mass index was the main predictive factor of NAFLD status in comparison with the other available parameters. A binomial logistic regression of NAFLD association with obesity, dyslipidaemia, and known history of hypertension showed statistical significance for obesity (Table 3). Further multinomial logistic regression of the association between the severity of NAFLD and obesity, dyslipidaemia, and known history of hypertension also pointed to obesity as the predictor of severe steatosis (Table 4). Additional linear regression analysis between obesity and NAFLD also showed a strong correlation, as shown in Figure 2.

## 4. Discussion

This study aimed at investigating the prevalence of, severity of, and contributing factors to NAFLD in people with T2DM in Ghana, using transient elastography. The average CAP score for the entire T2DM study population was 241 dB/m, which is approximately 30% steatosis. The average CAP score of T2DM patients with NAFLD was 278 dB/m, compared to 213 dB/m in T2DM without NAFLD. Fibrosis scores of 7 kPa and higher were found in 11.6% of the NAFLD cohort, none of whom had advanced fibrosis. Overweight/obesity was the main contributing factor to NAFLD, which was also evident in the association of waist circumference with NAFLD. Remarkably, a third of normal weight individuals with T2DM also had NAFLD.

For the first time, this study reports a 51.4% prevalence of NAFLD in a T2DM population from Ghana. This study is one of the few that reports the prevalence of co-existent NAFLD and T2DM in African populations. It suggests that the prevalence of NAFLD in persons with T2DM is higher than the general perception of low NAFLD rates in Africa. The initial Nigerian study in 2011 by Onyekwere et al. [18], which was followed by another study by Olusanya et al. [19] in 2016, and the recent one by Afolabi et al. [20] in 2018 reported rates of 9.5%, 16.7%, and 68.8%, respectively. This suggests that the prevalence of coexisting NAFLD and T2DM is steadily rising in sub-Saharan African populations. A related Ethiopian study by Zawdie et al. [21] similarly reported a prevalence of 73%. However, these earlier studies all utilized B-mode ultrasound in detecting NAFLD and were conducted in tertiary hospitals. This present study, unlike previous ones, was conducted at local community hospitals that primarily provide primary health care. It also used transient elastography, an ultrasound-based imaging technique with better sensitivity than B-mode ultrasound for detecting mild steatosis [22].

Transient elastography estimates liver fibrosis and steatosis with acceptable accuracy [23]. It has been well validated for liver stiffness measurement (LSM) to detect advanced fibrosis in different kinds of liver disease. The controlled attenuation parameter (CAP) in the transient elastography device is used for assessing steatosis noninvasively, allowing simultaneous assessment of fibrosis and steatosis [24]. However, there is a paucity of data from Africa on the assessment of NAFLD in persons with T2DM by transient elastography.

Using transient elastography, this study found that the average CAP score of the T2DM population was 241 dB/m, which is only 7 dB/m below the threshold for diagnosing NAFLD. It indicates that there is some increase in fat accumulation in nearly all T2DM cases in this study and those below the threshold are likely also at risk for NAFLD. The increase in liver fat is an adaptive response to metabolic stress to counteract the lipotoxicity of free fatty acids [25]. In T2DM, although there is an increase in liver lipogenesis [25], fatty acid oxidation is reduced [26], and triglyceride secretion via very low-density lipoprotein (VLDL) is inhibited [27]. Further, peripheral insulin resistance increases fatty acid release from adipose tissue [28], and insulin resistance increases hepatic fatty acid uptake [29]. Therefore, it is not uncommon for NAFLD and T2D to co-exist.

The majority of the Ghanaian T2DM cohort had NAFLD (51.4%), with an average CAP score of 278 dB/m, indicating moderate-to-severe steatosis. In terms of severity, this is only 2 dB/m shy of the severe range of 280 dB/m or higher. Previous research did not assess the severity of NAFLD in people with T2DM in Africa; therefore this study fills a gap in our knowledge of the condition in Africa. The progression of NAFLD to NASH and cirrhosis is strongly associated with T2DM [30]. It is estimated that 10–20% of T2DM patients with NAFLD may progress to NASH, increasing their risk of developing advanced fibrosis or cirrhosis [31]. T2DM is a risk factor for significant liver fibrosis. It has also been reported that about 17.7% of T2DM patients showed evidence of advanced liver fibrosis measured by transient elastography [32,33]. In the present study, 11.6% (13/112) of the NAFLD and T2DM group had fibrosis of 7.0 kpa or higher, indicating that they were at a higher risk for NASH. This percentage agrees with previous studies from western nations [27], which report a range of 10–20%.

It is important to note that the primary factor independently associated with NAFLD in T2DM was being overweight/obese. The average BMI of T2DM patients with and without NAFLD was 28.7 kg/m^2^ and 25.2 kg/m^2^, respectively, which are both considered overweight. The waist circumference, which showed a statistically significant association with NAFLD in this study, also confirmed the impact of abdominal fat distribution. The prevalence of NAFLD increases concurrently with the incidence of obesity, metabolic syndrome, and T2DM. An important pathophysiological factor in metabolic disease is the distribution of fat, and abdominal obesity may not have the same effects as evenly distributed fat. Since both groups in our study were overweight, our findings suggest that higher waist circumference may have accounted for the significant difference between the T2DM group with NAFLD and the group without NAFLD in our study. The lower waist circumference in the T2DM group without NAFLD may also be the primary reason for their average CAP score of 213 dB/m being less than the NAFLD threshold of 248 dB/m. If both groups successfully lose weight, their CAP scores may reduce, which could prevent NAFLD or minimise its severity.

This study was limited by the fact that transient elastography is known to have a specificity of 82.2% [22], which might have affected the interpretation of a negligible percentage of the population. In addition, the scope of this cross-sectional study did not include the diabetic medications of the participants, due to a lack of uniformity in medication adherence rates in Ghana [34]. Recent biochemical data of less than 3 months were also not readily available in all cases because of the affordability of diagnostic tests for diabetes in low-income settings [35]. With the use of a cross-sectional study design, this study was unable to determine cause–effect relationships and hence a longitudinal cohort study is recommended for future studies. Nonetheless, this study successfully achieved its purpose of using transient elastography to determine the prevalence and severity of NAFLD.

## 5. Conclusions

In summary, this study found that the prevalence of NAFLD in people with T2DM in Ghana is 51.4%, which is comparable to the reported global prevalence. In the population with co-existing NAFLD and T2DM, the CAP score obtained by transient elastography indicates moderate-to-severe steatosis. Being overweight/obese was the major independent contributing factor, which was also evidenced by a significant association with increased waist circumference. To prevent or minimize NAFLD, weight loss strategies should be encouraged in this population, which may be addressed in future studies.

## Figures and Tables

**Figure 1 jcm-12-03741-f001:**
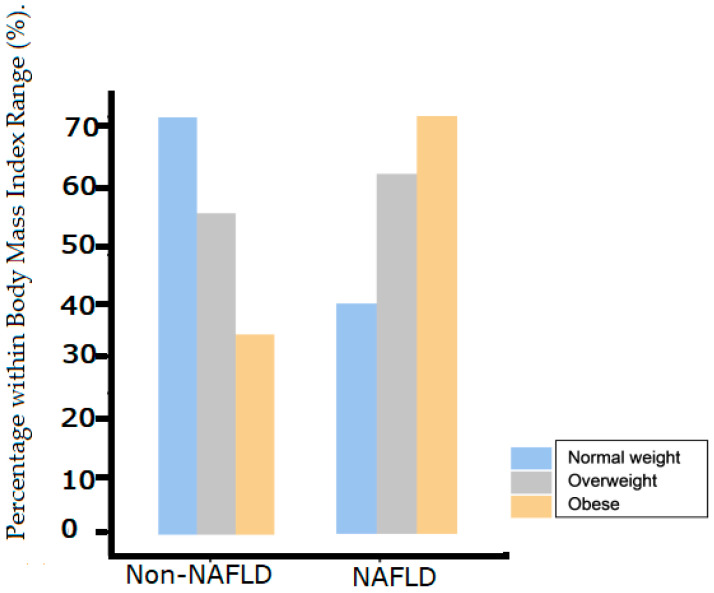
Body mass index proportions of NAFLD and Non-NAFLD groups.

**Figure 2 jcm-12-03741-f002:**
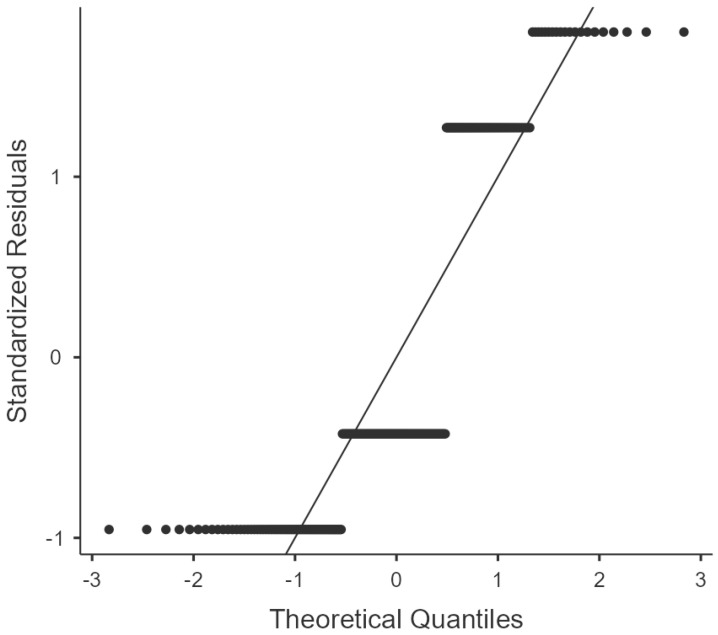
Linear logistic regression of obesity and NAFLD.

**Table 1 jcm-12-03741-t001:** Demographic, anthropometric, lifestyle, and clinical characteristics of T2DM patients with or without NAFLD.

		T2DM		
Variables	Total (*n* = 218)	Non-NAFLD (*n* = 106)	NAFLD (*n* = 112)	*p*-Value
Age category (years)				0.162
20–29	1 (0.5)	1 (100)	-	
30–39	9 (4.1)	5 (55.6)	4 (44.4)	
40–49	31 (14.2)	14 (45.2)	16 (54.8)	
50–59	62 (28.4)	25 (40.3)	37 (59.7)	
60–69	75 (34.4)	36 (48.0)	39 (52.0)	
70–79	30 (13.8)	17 (56.7)	13 (43.3)	
80–89	10 (4.6)	7 (70)	3 (30)	
Sex				0.064
Male	48 (22.0)	29 (60.4)	19 (39.6)	
Female	170 (78.0)	77 (45.3)	93 (54.7)	
Duration of T2DM				0.344
Less than a year	28 (12.8)	12 (11.3)	16 (14.3)	
1–5 years	90 (41.2)	40 (37.7)	50 (44.6)	
More than 5 years	100 (45.8)	54 (51.0)	46 (41.1)	
Smoking, current n (%)	3 (1.6)	2 (66.7)	1 (33.3)	0.608
Daily physical activity				0.756
<30 min	116 (53.2)	55 (51.9)	61 (54.5)	
≥30 min	102 (46.8)	51 (48.1)	51 (45.5)	
Hypertension, history, n (%)	151 (69.2)	71 (67.0)	80 (71.4)	0.542
Dyslipidaemia, history, n (%)	35 (16.1)	12 (11.3)	23 (20.5)	0.064
Significant fibrosis, n (%)	18 (8.3)	5 (5.7)	13 (11.6)	0.266
F1 (7–7.8 kPa)	9 (50.0)	2 (33.4)	7 (53.8)	
F2 (7.9–8.8 kPa)	8 (44.4)	4 (66.6)	4 (30.8)	
F3 (8.9 to <11.7 kPa)	1 (5.6)	0 (0.0)	2 (15.4)	
F4 (≥11.7 kPa)	0 (0.0)	0 (0.0)	0 (0.0)	
BMI group (kg/m^2^)				**<0.0001**
Normal weight (<25)	77 (35.3)	51 (66.2)	26 (33.8)	
Overweight (25–29.9)	73 (33.5)	34 (46.6)	39 (53.4)	
Obese (≥30)	68 (31.2)	20 (29.4)	48 (70.6)	
WC (cm)	101.0 (96.0–109.8)	98.0 (91.2–105.0)	106.0 (98.0–113.8)	**0.001**
HC (cm)	104.0 (96.0–111.8)	100.5 (93.3–108.0)	107.0 (99.0–115.0)	**0.003**
WHR	0.98 (0.94–1.03)	0.97 (0.93–1.03)	0.99 (0.94–1.03)	0.340
WHtR	0.64 (0.59–0.69)	0.62 (0.56–0.67)	0.66 (0.61–0.71)	**0.001**
SBP (mmHg)	136.0 (120.0–150.0)	131.0 (120.0–155.0)	136.0 (120.3–147.8)	0.857
DBP (mmHg)	80.0 (70.0–87.0)	79.5 (70.0–87.0)	80.0 (71.0–87.8)	0.996
CAP score (dB/m)	241.0 (214.0–281.0)	213.0 (188.8–224.3)	278.0 (257.3–316.8)	**<0.0001**

Continuous data are presented as median (interquartile range); compared using Mann–Whitney U-test. Categorical data are presented as number (%); compared using a chi-square or Fisher exact test. *p*-value < 0.05 was considered statistically significant. Bold values indicate significant values. N: number, T2DM: type 2 diabetes mellitus, BMI: body mass index, WC: waist circumference, HC: hip circumference, WHR: waist-to-hip ratio, WHtR: waist-to-height ratio, SBP: systolic blood pressure, DBP: diastolic blood pressure, CAP: controlled attenuation parameter.

**Table 2 jcm-12-03741-t002:** Multinomial logistic regression of the predictive factors of NAFLD. Bold values indicate significant values. SE: standard error.

NAFLD STATUS	Predictor	Estimate	SE	Z	*p*
S1–S0	Intercept	−5.176	2.017	−2.565	0.010
	Duration of diabetes	0.037	0.350	0.106	0.915
	Hypertension	−1.088	0.498	−2.184	0.029
	Daily physical activity	0.224	0.444	0.505	0.614
	Smoking	0.066	0.799	0.083	0.934
	Age	0.017	0.023	0.751	0.453
	BMI (kg/m^2^)	0.124	0.042	2.957	**0.003**
S2–S0	Intercept	−5.823	2.178	−2.674	0.007
	Duration of diabetes	−0.842	0.350	−2.405	0.016
	Hypertension	0.069	0.550	0.127	0.899
	Daily physical activity (2)	0.327	0.462	0.708	0.479
	Smoking	−0.446	0.914	−0.488	0.625
	Age	0.036	0.024	1.494	0.135
	BMI (kg/m^2^)	0.145	0.043	3.383	**<0.001**
S3–S0	Intercept	−5.957	2.078	−2.866	0.004
	Duration of diabetes	−0.224	0.351	−0.637	0.524
	Hypertension	0.863	0.599	1.440	0.150
	Daily physical activity (2)	−0.064	0.441	−0.147	0.883
	Smoking	−1.098	1.163	−0.944	0.345
	Age	0.002	0.023	0.117	0.907
	BMI (kg/m^2^)	0.177	0.041	4.307	**<0.001**

**Table 3 jcm-12-03741-t003:** Binomial logistic regression of NAFLD association with obesity, dyslipidaemia, and hypertension. SE: Standard Error.

Model Coefficients—NAFLD STATUS (YES/NO)						95% Confidence Interval	
Predictor	Estimate	SE	Z	*p*	Odds Ratio	Lower	Upper
Intercept	−0.496	0.309	−1.604	0.109	0.609	0.332	1.12
Obesity	1.379	0.389	3.546	<0.001	3.970	1.853	8.51
Dyslipidaemia	0.548	0.421	1.301	0.193	1.730	0.758	3.95
Hypertension	0.101	0.354	0.285	0.776	1.106	0.553	2.21

**Table 4 jcm-12-03741-t004:** Multinomial logistic regression of the association between the severity of NAFLD and obesity, dyslipidaemia and hypertension. SE: Standard Error.

Model Coefficients—NAFLD STATUS							95% Confidence Interval	
	Predictor	Estimate	SE	Z	*p*	Ratio	Lower	Upper
S1–S0	Intercept	−0.994	0.369	−2.690	0.007	0.370	0.179	0.763
	Hypertension	−0.839	0.462	−1.815	0.069	0.432	0.175	1.069
	Obesity	0.852	0.531	1.604	0.109	2.345	0.828	6.640
	Dyslipidaemia	0.983	0.519	1.893	0.058	2.672	0.966	7.393
S2–S0	Intercept	−2.074	0.533	−3.889	<0.001	0.126	0.044	0.357
	Hypertension	0.690	0.569	1.213	0.225	1.994	0.653	6.087
	Obesity	1.386	0.531	2.614	0.009	4.001	1.415	11.318
	Dyslipidaemia	−0.021	0.653	−0.032	0.974	0.979	0.272	3.521
S3–S0	Intercept	−2.186	0.520	−4.200	<0.001	0.112	0.040	0.312
	Hypertension	0.827	0.537	1.542	0.123	2.289	0.799	6.554
	Obesity	1.801	0.485	3.713	<0.001	6.060	2.343	15.684
	Dyslipidaemia	0.433	0.553	0.782	0.434	1.542	0.521	4.561

## Data Availability

The data presented in this study are available upon request from the corresponding author.

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
