# Peer review of "Non-Alcoholic Fatty Liver Disease and Liver Fibrosis in Persons with Type 2 Diabetes Mellitus in Ghana: A Study of Prevalence, Severity, and Contributing Factors Using Transient Elastography"

_jcm, 2023, doi:10.3390/jcm12113741_

Round 1
Reviewer 1 Report
This is an interesting research, revealing new information regarding prevalence of NAFLD in adults with TD2 of African origin. Some questions:
1) why you haven't included information from biochemical data of your patients, ie liver enzymes, HOMA, etc
2) were the TD2 patients under any treatment?
3) from the results section, it is obvious that the majority of patients were young (20-40 years old) and female. Could you make some comments regarding these findings in the discussion section?
Author Response
Dear Reviewer,
Thank you so much for reviewing this manuscript and providing us with constructive comments which has now improved the quality of our work. Below are our responses to your comments.
This is an interesting research, revealing new information regarding prevalence of NAFLD in adults with TD2 of African origin. Some questions:
1) why you haven't included information from biochemical data of your patients, ie liver enzymes, HOMA, etc RESPONSE: Yes, we have stated the reason at line 269-272
2) were the TD2 patients under any treatment? RESPONSE: We have stated the response at lines 269 - 272
3) from the results section, it is obvious that the majority of patients were young (20-40 years old) and female. Could you make some comments regarding these findings in the discussion section? RESPONSE: We have made revisions in table 1 and lines 132-134 to address the miscommunication.

Reviewer 2 Report
I have the following comments from the manuscript by Wiafe Y. A. and colleagues titled “Non-Alcoholic Fatty Liver Disease and Liver Fibrosis in Persons with Type 2 Diabetes Mellitus in Ghana: A Study of Prevalence, Severity, and Contributing Factors Using Transient Elastography”. The paper is interesting; however, the authors should address the following concerns.
-In the abstract section, T2DM should define correctly, including NAFLD. In addition, the main results described in this section are difficult to understand because the groups analyzed were not described. Please define de groups and add the number of patients. The results of fibrosis should also be included.
-In the introduction section, please include diabetes prevalence in Africa, specifically in Ghana.
-There are some typos in the manuscript.
-In materials and methods, the authors should describe this section by subtitles according to the experimental design. In general, this section should describe the methods in more detail. Please include antidiabetic drugs that patients were taking, how the lipid profile was evaluated and which ones were considered for dyslipidemia? Describe if blood samples were collected in tubes containing ethylene diamine tetraacetic acid (EDTA), including samples processing.
-Why in table 1 the percentages were calculated according to the number patients by parameter and not by number of group? Moreover, the percentages should check again to confirm they are correct. The table 4 was not described in the manuscript.
-In discussion section, please include that diabetes is a risk factor for significant liver fibrosis.
-In the following phrase (page 7, line 204): “The initial Nigerian study in 2011 by Onyekwere et al [15],” The reference is not correct, please add the appropriate one.
-Clarify how the percentage mentioned in the following sentence was calculated? (page 8, line 240): “In the present study, 11.6% of the NAFLD and T2DM group had fibrosis of 7.0 kpa or 240 higher,”
There are some typos in the manuscript.
Author Response
Dear Reviewer,
Thank you so much for reviewing this manuscript and providing us with constructive comments which has now improved the quality of our work. Below are our responses to your comments.
- -In the abstract section, T2DM should define correctly, including NAFLD. In addition, the main results described in this section are difficult to understand because the groups analyzed were not described. RESPONSE: Addressed at lines 18 -20, 31
- Please define de groups and add the number of patients. RESPONSE: Addressed at lines 30-31
- The results of fibrosis should also be included. RESPONSE: Addressed at lines 30-31
- -In the introduction section, please include diabetes prevalence in Africa, specifically in Ghana. RESPONSE: addressed at lines 132-134
-There are some typos in the manuscript. RESPONSE: addressed
-In materials and methods, the authors should describe this section by subtitles according to the experimental design. In general, this section should describe the methods in more detail. Please include antidiabetic drugs that patients were taking, how the lipid profile was evaluated and which ones were considered for dyslipidemia? Describe if blood samples were collected in tubes containing ethylene diamine tetraacetic acid (EDTA), including samples processing. RESPONSE: Please, respectfully, this was not applicable on this occasion. Kindly check response at lines 269-272.
-Why in table 1 the percentages were calculated according to the number patients by parameter and not by number of group? Moreover, the percentages should check again to confirm they are correct. The table 4 was not described in the manuscript. RESPONSE: Checked
-In discussion section, please include that diabetes is a risk factor for significant liver fibrosis. RESPONSE: Addressed at line 246
-In the following phrase (page 7, line 204): “The initial Nigerian study in 2011 by Onyekwere et al [15],” The reference is not correct, please add the appropriate one. RESPONSE: Thank you. It is now addressed as reference 18
-Clarify how the percentage mentioned in the following sentence was calculated? (page 8, line 240): “In the present study, 11.6% of the NAFLD and T2DM group had fibrosis of 7.0 kpa or 240 higher,” RESPONSE: Thank you. It is now addressed in line 248
- Comments on the Quality of English Language
There are some typos in the manuscript. RESPONSE: Addressed

Round 2
Reviewer 2 Report
I have no comments.
English language is fine